

# Reporting and interpreting non-significant results in animal cognition research

Benjamin G. Farrar[1,2], Alizée Vernouillet[3], Elias Garcia-Pelegrin[1,4], Edward W. Legg[5,6,7], Katharina F. Brecht[8], Poppy J. Lambert[9], Mahmoud Elsherif[10,11], Shannon Francis[12], Laurie O'Neill[12], Nicola S. Clayton[1] and Ljerka Ostojić[5,6,7]

[1] Department of Psychology, University of Cambridge, Cambridge, United Kingdom
[2] Institute for Globally Distributed Open Research and Education (IGDORE), Cambridge, United Kingdom
[3] Department of Experimental Psychology, Universiteit Gent, Gent, Belgium
[4] Department of Psychology, National University of Singapore, Singapore, Singapore
[5] Department of Psychology, Faculty of Humanities and Social Sciences, University of Rijeka, Rijeka, Croatia
[6] Division of Cognitive Sciences, University of Rijeka, Rijeka, Croatia
[7] Centre for Mind and Behaviour, University of Rijeka, Rijeka, Croatia
[8] Institute for Neurobiology, University of Tuebingen, Tuebingen, Germany
[9] Messerli Research Insititute, University of Vienna, Vienna, Austria
[10] Department of Psychology, University of Birmingham, Birmingham, United Kingdom
[11] University of Leicester, Leicester, United Kingdom
[12] Comparative Cognition Research Group, Max Planck Institute for Ornithology, Seewiesen, Germany

Corresponding authors
Benjamin G. Farrar,
farrarbg@gmail.com
Ljerka Ostojić, lj.ostojic@uniri.hr

## ABSTRACT

How statistically non-significant results are reported and interpreted following null hypothesis significance testing is often criticized. This issue is important for animal cognition research because studies in the field are often underpowered to detect theoretically meaningful effect sizes, *i.e.*, often produce non-significant $p$-values even when the null hypothesis is incorrect. Thus, we manually extracted and classified how researchers report and interpret non-significant $p$-values and examined the $p$-value distribution of these non-significant results across published articles in animal cognition and related fields. We found a large amount of heterogeneity in how researchers report statistically non-significant $p$-values in the result sections of articles, and how they interpret them in the titles and abstracts. Reporting of the non-significant results as "No Effect" was common in the titles (84%), abstracts (64%), and results sections (41%) of papers, whereas reporting of the results as "Non-Significant" was less common in the titles (0%) and abstracts (26%), but was present in the results (52%). Discussions of effect sizes were rare (<5% of articles). A $p$-value distribution analysis was consistent with research being performed with low power of statistical tests to detect effect sizes of interest. These findings suggest that researchers in animal cognition should pay close attention to the evidence used to support claims of absence of effects in the literature, and—in their own work—report statistically non-significant results clearly and formally correct, as well as use more formal methods of assessing evidence against theoretical predictions.

## INTRODUCTION

Null hypothesis significance testing (NHST) is a primary method of statistical analysis in animal cognition research. However, when NHST produces results that are not statistically significant, these are often difficult to interpret. If researchers test null hypotheses of zero effect (*i.e.,* there are no differences between groups or conditions), a non-significant result could result from a lack of any effect in the population (a true negative), or a failure to detect some true difference (a false negative). While current guidance encourages researchers to design studies with high statistical power to detect theoretically interesting effect sizes (*Lakens, 2017*; *Lakens, 2021*)—which can provide context for negative results—power analyses appear infrequent (*Fritz, Scherndl & Kühberger, 2013*). Hence, how statistically non-significant results are reported and interpreted within the NHST framework has been criticized on several grounds (*Fiedler, Kutzner & Krueger, 2012*; *Gigerenzer, Krauss & Vitouch, 2004*; *Lambdin, 2012*; *Vadillo, Konstantinidis & Shanks, 2016*). The most prominent criticism is that researchers often misreport a non-significant difference between test conditions or groups as if the result in each of the conditions or groups was exactly the same, and/or researchers misinterpret a non-significant results as evidence of the absence of an effect in regard to a substantive claim (*Aczel et al., 2018*; *Fidler et al., 2006*; *Hoekstra et al., 2006*). This misreporting or misinterpretation may even occur when the null hypothesis being considered is very likely to be incorrect (*Cohen, 1994*; *Gelman & Carlin, 2014*). Given these concerns, this study explored how animal cognition researchers report and interpret statistically non-significant results using a manually extracted dataset of negative claims following NHST from over 200 articles.

When using NHST, researchers attempt to reject a statistical model (the null hypothesis) with their data while controlling the rate at which they will make false-positive decisions in the long-term (*Neyman & Pearson, 1933*). Most often, this statistical null is that there is absolutely no difference between two groups or conditions (for example a mean difference of 0 for a *t*-test; 'nil' hypothesis; *Cohen, 1994*), or, in the case of a one-tailed test, that the difference will not be zero *or* that it will be *not* in a certain direction, *i.e.,* researchers make a directional prediction for their alternative hypothesis. A statistical test then produces a *p*-value, *i.e.,* the probability of observing the researchers' data or more extreme data if the null hypothesis and all its assumptions were true, $\Pr(d(X) \geq d(x_0); H_0)$. If the *p*-value is lower than a pre-specified threshold (the $\alpha$ level), the statistical null hypothesis ($H_0$) is rejected in favor of an alternative hypothesis (*Neyman & Pearson, 1933*), whereas if the *p*-value is larger than the pre-specified threshold, $H_0$ should not be rejected. However, how researchers should behave towards their null and alternative hypotheses following a non-significant result has been a continued locus of criticism of NHST (*Lambdin, 2012*). Formally, researchers *can* make statements about the long-run error probabilities of their test procedures. For example, with an $\alpha$ level of .05 and if no $\alpha$-inflating research practices

were used (*Simmons, Nelson & Simonsohn, 2016*), they can say that in the long run they would not reject $H_0$ more than 5% of the time, if $H_0$ were true. Similarly, if the design of the study is such that the statistical test had 90% power to detect the smallest effect size of interest, in the long run the researchers would only fail to reject $H_0$ 10% of the time, if the smallest effect size of interest did exist in the population.

Without performing further analyses, it can be an error to conclude that there is evidence in favor of the null hypothesis following a non-significant result. The arbitrary nature of the $\alpha$ level highlights this: as an example, let us assume that we calculate a *p*-value of 0.08 with an $\alpha$ level of .05. By not rejecting $H_0$ in this instance, we can say that in the long run we would not reject $H_0$ more than 5% of the time, if it were true. However, if we had chosen an $\alpha$ level of .10 instead, we would have rejected $H_0$. Clearly, then, the *p*-value when using NHST is not a direct indication of the strength of evidence for or against $H_0$, but must be interpreted relative to error rates and alternative hypotheses (*Lakens et al., 2018*). However, despite the *p*-value not being the probability of the null hypothesis being true, survey studies suggest researchers do interpret *p*-values in such a way (*e.g., Goodman, 2008*). Moreover, scientists often misreport non-significant results as evidence of absence of a difference between groups or conditions or evidence of no effect when this inference is not necessarily warranted (*Aczel et al., 2018*; *Fidler et al., 2006*; *Hoekstra et al., 2006*). Such an error might be especially important in animal cognition research, in which a combination of small sample sizes and low trial number may limit the ability of researchers to design studies and statistical test combinations with high power of statistical tests to detect the minimum effect size of theoretical interest (*Farrar, Boeckle & Clayton, 2020*).

While 'accepting the null' may be an error, just how severe an error it is requires discussing. Just because a researcher might report the results of significance tests incorrectly, this does not mean that they themselves, or their readers, necessarily interpreted the significance test incorrectly. In their 1933 article, Neyman and Pearson often discussed 'accepting $H_0$' following a result that was not statistically significant (*Neyman & Pearson, 1933*). In fact, as *Mayo* (*2018*, p. 135) writes, Neyman used the term 'acceptance' as shorthand, and even preferred the phrase "No evidence against [the null hypothesis] is found" to "Do not reject [the null hypothesis]" (*Neyman, 1976*, postscript, p. 749). If scientists equate phrases such as "there were no differences between conditions ($p > 0.05$)" with "there was no statistically significant difference between the conditions", then the "serious mistake" of accepting the null becomes an issue of precision in language, rather than an egregious error.

The aim of this study was to explore how authors in fields related to animal cognition report and interpret statistically non-significant results by building on the methods used in similar studies in psychology and conservation biology (*Aczel et al., 2018*; *Fidler et al., 2006*; *Hoekstra et al., 2006*). This is an important step towards (i) identifying how often conclusions in animal cognition might be the result of NHST misreporting or misinterpretation, and (ii) highlighting areas in which animal cognition researchers can improve their statistical inferences and statistical reporting.

**Table 1  Sources of articles containing negative results in their abstracts.**

| Source | N articles |
| --- | --- |
| *Animal Behaviour* | 13 |
| *Animal Behavior and Cognition* | 14 |
| *Animal Cognition* | 17 |
| *Animals* | 15 |
| *Applied Animal Behaviour Science* | 15 |
| *Behaviour* | 14 |
| *Behavioural Processes* | 15 |
| *Ethology* | 16 |
| *Frontiers in Psychology: Comparative Psychology* | 14 |
| *Frontiers in Veterinary Science: Animal Behaviour and Welfare* | 15 |
| *International Journal of Comparative Psychology* | 13 |
| *Journal of Applied Animal Welfare Science* | 15 |
| *Journal of Comparative Psychology* | 15 |
| *Journal of Ethology* | 15 |
| *Journal of Experimental Psychology: Animal Learning and Cognition* | 16 |
| *Journal of Zoo and Aquarium Research* | 15 |
| *Learning and Behavior* | 15 |
| *PeerJ: Animal Behaviour* | 15 |
| *bioRxiv: Animal Behaviour and Cognition* | 14 |
| *PCI: Animal Science* | 2 |

# MATERIALS & METHODS

## Data extraction and classification

We manually extracted data from a total of 20 sources, comprising 18 peer-reviewed journals in animal cognition, behavior, and welfare, one pre-print server, and articles recommended through Peer Community in Animal Science. The 20 sources are detailed in Table 1. Sources were screened backwards from March 2021, until 15 articles were identified that contained negative statements in titles or abstracts that corresponded to statistically non-significant results from null-hypothesis significance tests in the article, or until all articles in that source had been viewed. If coders were uncertain about whether an article should be included, they continued until they had 15 articles that they were confident with, explaining why three journals had 16 or 17 articles extracted.

Nine of the authors (author acronyms: BGF, AV, KB, EGP, LoN, PL, SF, EL, and ME) performed the coding and were each assigned two journals, except BGF who conducted the coding for four journals. Each coder screened the abstracts of each article of their assigned journals and identified any statements that was either, (i) reporting the results of a statistically non-significant hypothesis test (what we referred to as 'sample claim' in the Coding guidelines, https://osf.io/84puf/, as *Aczel et al., 2018*) and/or (ii) interpreted a statistically non-significant result in relation to a substantive claim (what we referred to as 'population claim' in the Coding guidelines, https://osf.io/84puf/, as *Aczel et al., 2018*). If at
least one of these statements was present, the coder recorded the article's information (title, first author, journal, and year) and the statement(s) in question. For articles with multiple negative statements within each of the categories (reporting of results, interpretation of results in relation to a substantive claim), the coder recorded the statement that they thought was most clearly related to the paper's main claim, such that for each article, we had a maximum of one result statement and one substantive interpretation statement. Next, the coder verified that the statements were based on results from NHST. If verified, the coder then extracted the text of the NHST that corresponded to the abstract claim from the results section of the manuscript, including the associated *p*-value. If there was more than one corresponding statistical test within an experiment, the coder extracted the test result that they thought was most relevant to the claim. If the abstract claim was equally supported by multiple studies or experiments, the coder extracted the information from the first study or experiment presented.

After the title, abstract claim(s) (reporting of results and interpretations of the results in relation to a substantive claim), result text and *p*-value had been extracted, the coder categorized how each statement was phrased. Through piloting, discussion and from looking at previous studies (*Aczel et al., 2018*; *Fidler et al., 2006*; *Hoekstra et al., 2006*), we developed three categories. For the statements that reported the results in the title and abstract and for the result text, these were: (1) "Formally Correct, *i.e.,* Non-Significant" statements that either there was no *significant* difference between testing groups or conditions, or words to that effect, or a correct directional statement (text slightly altered from original phrasing; for the original phrasing see our Coding Guidelines, https://osf.io/84puf/); (2) "No Effect" statements that there was not a difference between testing groups or conditions, when in fact there was—it was just not significant in the analysis (text slightly altered from original phrasing; for the original phrasing see our Coding Guidelines, https://osf.io/84puf/); (3) "Ambiguous, Similar or Small Effect Size" statements about the results that neither suggest that the testing groups or conditions were the same, nor that there was no significant difference between them (which were later split into "Ambiguous" and "Similar or Small Effect Size" categories; text slightly altered from original phrasing; for the original phrasing see our Coding Guidelines, https://osf.io/84puf/). In addition to these descriptions, we developed a table of hypothetical statements that are detailed in Table 2, which were available to the coders during the project.

Similarly, the title, if it contained a statement referring to a statistically non-significant result and interpretations of the results in relation to a substantive claim from the abstracts were categorized into three categories: (1) "Correct, *i.e.,* Justified": An interpretation that commented on statistical power, use of equivalence tests or otherwise a justification why a non-significant result suggests that there is no theoretically important difference in regards to the substantive claim, or that the study provides no strong evidence of a difference (text slightly altered from original phrasing; for the original phrasing see our Coding Guidelines, https://osf.io/84puf/), (2) "Caveated or Ambiguous": An interpretation of the non-significant results as suggesting/indicating etc. that X and Y do not differ, or showing that they are similar, and (3) "No Effect": An incorrect interpretation of the non-significant result as showing that X and Y do not differ in relation to a substantive

**Table 2** Example categorization of sample-level statements.

| Category | Non-Significant | No Effect | Ambiguous, Similar, or Small Effect Size |
|---|---|---|---|
| Description | Reports that there was no *significant* difference between two conditions, or words to that effect. | A statement that there was not a difference within the sample, when in fact there was—it was just not significant in their analysis. | A statement about the results that neither suggests they were the same, nor that there was no significant difference. |
| Examples | There was no significant/detectable difference between X and Y. We did not detect a difference between X and Y (or any other statement implying failing to find a signal within noise). We did not find a significant effect. X was not significantly related to Y. X did not perform significantly above chance. X performed significantly above chance, but Y did not. There were no significant differences between X and Y's performance. | There was no difference between X and Y. There was no effect. There was no evidence of an effect. There was no relationship between X and Y. We did not find/observe/see a difference between X and Y. We did not find an effect. We found no evidence of an effect. X performed at chance levels. | X and Y were similar. There was no large/clear difference between X and Y. There was no large effect of X on Y. |

claim (text slightly altered from original phrasing; for the original phrasing see our Coding Guidelines, https://osf.io/84puf/). In addition to these descriptions, we developed a table of hypothetical statements that are detailed in Table 3.

## Reliability and quality control

Twenty-four articles (8.5%) were double-blind coded to assess the likely reliability of our coding scheme, and all articles underwent a quality control procedure involving a second coder to identify any mistakes or inconsistencies in the extracted dataset before the data were used in any analysis.

### Double-blind extraction

To test the reliability of the coding scheme, BGF independently coded 24 articles, namely the first four articles from six randomly chosen journals, blind to the results of the original coders. From this, we computed inter-rate agreement for each variable that we extracted (title statement, reporting of results in abstract, interpretation of results in relation to a substantive claim in abstract, reporting of results in result section, $p$-value), including raw percentage agreement, Cohen's kappa and Gwet's AC1.

### Quality control

To minimize mistakes in the dataset, all articles underwent the quality control procedure. Here, a second coder reviewed the data extracted from each article. Ten of the authors (BGF, AV, KB, EGP, LoN, PL, SF, EL, ME, and LO) served as second coders, and each was assigned one other coder's original set of articles to quality control. The quality controller verified (1) that a claim from the title/abstract has been extracted, (2) that any

**Table 3  Example categorization of population-level or title claims.**

| Category | Justified | Caveated, Ambiguous or Similar | No Effect |
|---|---|---|---|
| Description | Comments on statistical power, uses equivalence tests or otherwise justifies why a non-significant result suggests that there is no theoretically important difference in the population, or that the study provides no strong evidence of a difference. | Interprets the non-significant results as suggesting/indicating etc. that X and Y do not differ in the population, or are similar. | Interprets the non-significant result as showing that X and Y do not differ in the population. |
| Examples | | …suggesting that X is not related to Y. | …meaning that X is not related to Y. |
| | | …indicating that X is not related to Y. | …showing that X is not related to Y. |
| | Because the test was high-powered to detect a meaningful difference, this non-significant result suggests that A is not related to Y in a theoretically important way. | | There is no difference between X and Y. |
| | | …suggesting/indicating that there is no difference between X and Y. | X and Y do not differ. |
| | | …suggesting that X has not changed Y. | X and Y are similar. |
| | | | X and Y are the same (show the same effect, etc.). |
| | In addition to being not statistically different to each other, X and Y were also statistically equivalent (if a frequentist equivalence or non-inferiority test was performed), suggesting that X is not meaningfully related to Y. | Our results provide no strong evidence that X and Y are different. | X does not change Y. |
| | | …suggesting that X and Y are similar. | Our results provide no evidence that X and Y are different. |

claim extracted referred to a statistically non-significant result, (3) that the result that was extracted corresponded to the claim that was extracted, and (4) that they agreed with the classification of each claim. If the quality controller identified a mistake, they classified this as a major disagreement, whereas if the quality controller disagreed but was uncertain about this judgment, for example in the case of borderline claims, they classified this as a minor disagreement. BGF reviewed all disagreements and made a final decision on what entered the final dataset, returning to the original article if necessary.

## Analysis

We present the percentage of claims in each category across the extracted variables (title statements, statements reporting non-significant results in the abstract, statements interpreting the results in relation to a substantive claim in abstracts, and reporting of the statistical result in the results section). To illustrate the types of claims placed in each category, examples that we felt were particularly representative of each are provided in tables. In addition, every classification can also be viewed in the open dataset at https://osf.io/84puf/. We used a Chi-squared test to test whether, if the results reported "No Effect", it was more likely that a "No Effect" interpretation would also be made in the abstract than when the results were correctly reported as "Non-Significant".

In addition to our primary descriptive analysis, we performed an exploratory analysis of the $p$-value distribution of the $p$-values associated with all extracted non-significant results. We used a two-sided Kolmogorov–Smirnov test to compare the observed distribution to

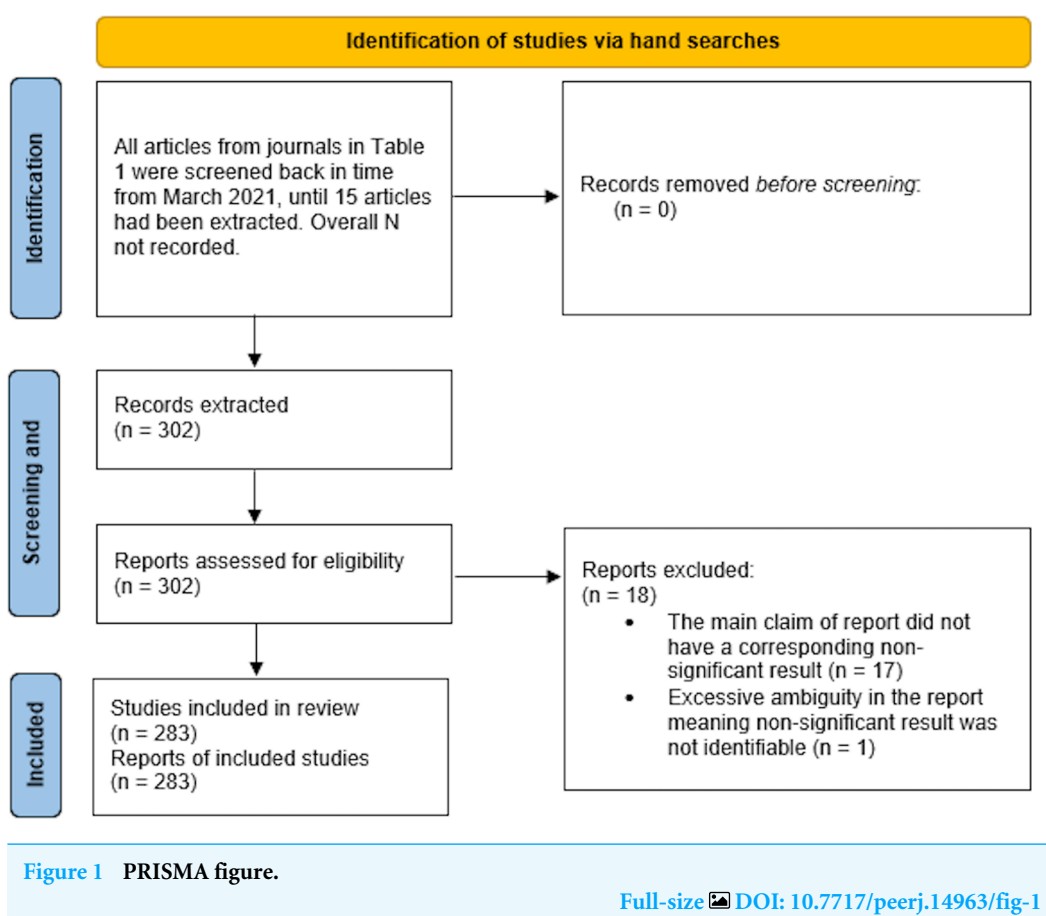

**Figure 1** PRISMA figure.

a uniform distribution, *i.e.,* the theoretical distribution that the *p*-values would have been samples from if the null hypothesis of no difference was true for all studies.

## RESULTS

We extracted data from 302 articles. Of these, 18 were excluded due to their identified claim having no corresponding negative result of NHST (*e.g.*, only descriptive statistics used, or only a Bayesian analysis performed) and one was excluded due to excessive ambiguity in how the results were described. This left a final sample of 283 articles for analysis (Fig. 1). The results of the Reliability and Quality Control coding can be found in the Supplemental Material.

### Title statements

Forty-four titles (19% of the total articles) were identified as containing statements resulting from non-significant results of NHSTs. Of these, 37 (84%) were classified as interpreting the non-significant result as evidence of no effect, whereas seven (16%) were classified as caveated claims or claims about testing groups or conditions being 'similar'. Table 4 provides examples of these claims.

**Table 4** Examples of claims in the titles of articles following non-significant NHST classified as "No Effect" and "Caveated or Similar".

**No Effect**

*N* = 37 (84%)

"Home range use in the West Australian seahorse Hippocampus subelongatus is influenced by sex and partner's home range but not by body size or paired status"

*Kvarnemo et al. (2021)*

"Delays to food-predictive stimuli do not affect suboptimal choice in rats."

*Cunningham & Shahan (2020)*

"Common Marmosets (*Callithrix jacchus*) Evaluate Third-Party Social Interactions of Human Actors But Japanese Monkeys (*Macaca fuscata*) Do Not"

*Kawai et al. (2019)*

**Caveated, Ambiguous, or Similar**

*N* = 7 (16%)

"Limited Evidence of Number-Space Mapping in Rhesus Monkeys (Macaca mulatta) and Capuchin Monkeys (*Sapajus apella*)"

*Beran et al. (2019)*

"Little Difference in Milk Fatty Acid and Terpene Composition Among Three Contrasting Dairy Breeds When Grazing a Biodiverse Mountain Pasture"

*Koczura et al. (2021)*

"The Equipment Used in the SF6 Technique to Estimate Methane Emissions Has No Major Effect on Dairy Cow Behavior"

*Pereira et al. (2021)*

## Abstract statements
### Reporting of results in abstracts

We extracted 278 claims that reported non-significant results of NHST. Of these, 174 (63%) were classified as claiming evidence of no effect, 71 (26%) as making formally correct statements that there were no statistically significant differences between groups or conditions, 17 (6%) as making claims about an effect being 'similar' between groups or conditions, or as describing a small effect size, and 16 (6%) were classified as ambiguous. Table 5 provides examples of these claims.

### Interpretations of results in abstracts

We extracted 63 statements that were interpretations of statistically non-significant results in relation to substantive claims. Of these, 45 (71%) were classified as caveated and 18 as claiming that there was no effect (29%). Table 6 provides examples of these claims.

## Result text

In the results sections, 276 non-significant results of NHST were coded. Of these, 140 (52%) were classified as reporting the results as "Non-Significant", 113 (41%) as reporting that there was "No Effect", 12 (4%) as reporting groups or conditions being "Similar", 10 (4%) were classified as "Ambiguous", and one (0.4%) as reporting a "trend" in the opposite direction to the prediction. Table 7 provides examples of the different types of result reporting.

**Table 5 Examples of claims about the sample in the abstracts of articles following non-significant NHST classified as "No Effect", "Similar or Small Effect Size", "Non-Significant" or "Ambiguous".**

**No Effect**

$N = 174$, 63%

"Levels of individuals sitting with their back to the window was unaffected by visitor number or noise."

*Hashmi & Sullivan (2020)*

"The groups did not differ in their ability to follow human signals"

*Lazarowski et al. (2020)*

**Similar or Small Effect Size**

$N = 17$, 6%

"Pair members demonstrated comparable responses towards a male 'intruder', as latency to respond and proximity scores were very similar between pair members in the majority of pairs examined"

*DeVries, Winters & Jawor (2020)*

"We found that individuals called back to sympatric and allopatric calls within similar amounts of time,"

*Wu et al. (2021)*

**Non-Significant**

$N = 71$, 26%

"Nutcrackers…did not significantly change their caching behaviour when observed by a pinyon jay."

*Vernouillet, Clary & Kelly (2021)*

"No significant correlations between degree of laterality and behavioral interest in the stimuli were found"

*Lilley, De Vere & Yeater (2020)*

**Ambiguous**

$N = 16$ (6%)

"We also found no conclusive evidence that either the visual or the vibratory sensory modalities are critical for prey capture."

*Meza, Elias & Rosenthal (2021)*

"No systematic variations on space allocation were observed in neither experiment"

*Ribes-Iñesta, Hernández & Serrano (2020)*

Notably, if a sentence reporting the results in the results section was classified as "No Effect", it was more likely that this statistical test would also be reported as "No Effect" in the abstract, compared to when the result was classified as "Non-Significant" ($\chi 2(1, N = 211) = 21.65, p < .0001$). Limiting the data to just those with responses in the abstract and results classified as "Non-Significant" or "No Effect", of the 92 statements in the results classified as "No Effect", 80 (87%) of the corresponding statements reporting the results in the abstract were classified as "No Effect". In contrast, of the 119 statements in the results classified as "Non-Significant", only 67 (56%) were reported as "No Effect" in the abstract. Nevertheless, the "No Effect" phrasing when reporting results in the abstracts was absolutely the most likely classifications for both "No Effect" and "Non-Significant" phrasings in the results section.

### *p*-value distributions

In total, 202 of the 283 articles reported exact *p*-values, with the other 81 reporting either inequalities or not reporting the *p*-values at all. Of these 202 *p*-values, four were below

**Table 6  Examples of claims about populations in the abstracts of articles following non-significant NHST classified as "No Effect" and "Caveated, Ambiguous or Similar".**

**No Effect**

$N = 18$ (29%)

"Partial rewarding does not improve training efficacy"

*Cimarelli et al. (2021)*

"Our findings show that *H. horridum* does not respond to hypoxic environments"

*Guadarrama et al. (2020)*

"Oviposition site choice is not by-product of escape response"

*Kawaguchi & Kuriwada (2020)*

**Caveated, Ambiguous, or Similar**

$N = 45$ (71%)

"These results suggest capuchin monkeys do not engage in indirect reciprocity"

*Schino et al. (2021)*

"These results suggest that shoal composition may not be an important driver of shoal choice in this system"

*Paijmans, Booth & Wong (2021)*

"…suggesting that size is not a determinant factor for feral horse society."

*Pinto & Hirata (2020)*

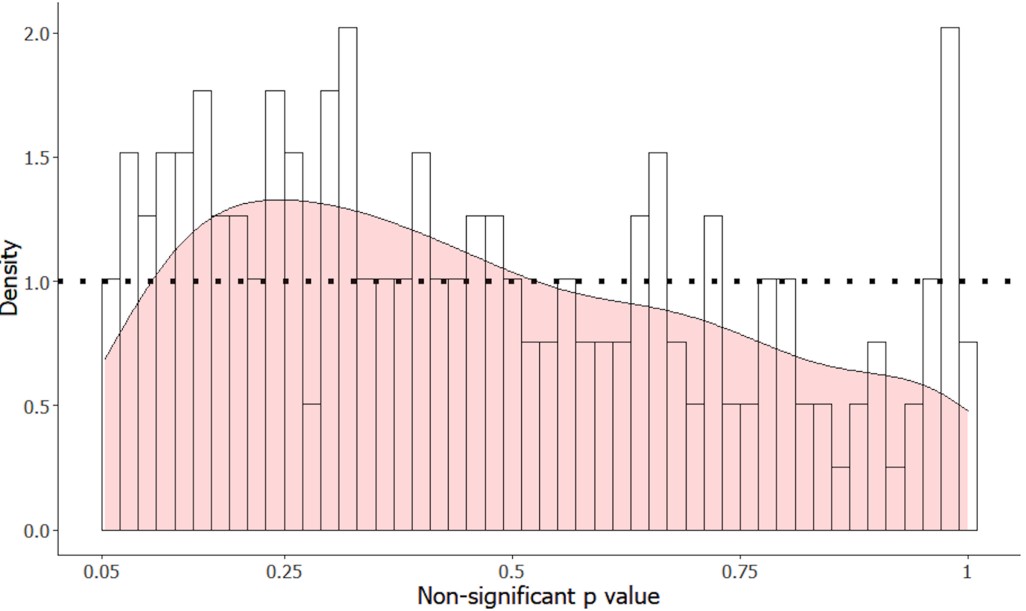

**Figure 2  Distribution of non-significant *p*-values from result sections of 198 articles in animal cognition and related fields, with a density distribution overlaid in pink.**  The dotted line shows the average density.

.05 and non-significant due to a lower $\alpha$ level. The distribution of the 198 non-significant *p*-values in the interval .05–1 is displayed in Fig. 2. This distribution significantly differs from a uniform distribution (two-sided Kolmogorov–Smirnov test, $D = 0.12$, $p = .0087$).

**Table 7** Examples of statement reporting the results in the results sections of articles using non-significant NHST classified as "No Effect", "Similar or Small Effect Size", "Non-Significant" or "Ambiguous".

**No Effect**

$N = 113$ (41%)

During farrowing, No Effect of the treatments was seen on the percentage of time spent (3.22% vs. 1.90%, $P = 0.372$) on the nest-building behaviour"

*Aparecida Martins et al. (2021)*

"There were no differences between treatments in the frequency or duration of birds flying between walls"

*Stevens et al. (2021)*

**Similar or Small Effect Size**

$N = 12$ (4%)

"The average time yaks spent grazing was similar among shrub coverage groups ( $P = 0.663$)"

*Yang et al. (2021)*

"The number of sessions required to reach criterion didn't reliably differ between groups"

*O'Donoghue, Broschard & Wasserman (2020)*

**Non-Significant**

$N = 140$ (52%)

"Comparing the pooled data of all crows, no significant increase in the number of mark-directed behaviors during the mirror mark condition was found compared with the no-mirror sham condition."

*Brecht, Müller & Nieder (2020)*

"There was no significant effect of removal type on changes in display strength in either dominant males or subordinate males."

*Piefke et al. (2021)*

**Ambiguous**

$N = 10$ (4%)

"As can be seen in Figure 1D, there was no difference in response rates after R and NR trials across days for rats under reward uncertainty." [where in Figure 1D the bars on the graph look almost identical]

*Anselme & Robinson (2019)*

"It showed that there was a significant main effect of session, but no main effect of CS"

*Harris & Bouton (2020)*

Figure 3 contrasts the distribution of Fig. 2 with the four simulated distributions of bodies of research performed where 80% of alternative hypotheses were correct, and studies had either 10, 33, 50 or 80% statistical power to detect the true effect size of H1 if it was true. Notably, $p$-values in the interval from .05 to .10 were underrepresented in the manually extracted data, making up only 5.6% of observations compared to 8.2% (10% power simulation), 15% (33% power simulation), 19% (50% power simulation), and 20% (80% power simulation). Similarly, very high $p$-values (.95 − 1.0) were overrepresented in our manual dataset (7.6% of observations, compared to 4.3%, 3.2%, 2.4% and 3.4% for the 10, 33, 50 and 80% power simulations respectively), which likely reflects either the use of multiple correction procedures, or small sample non-parametric statistics that produce non-uniform distributions under the null hypothesis.
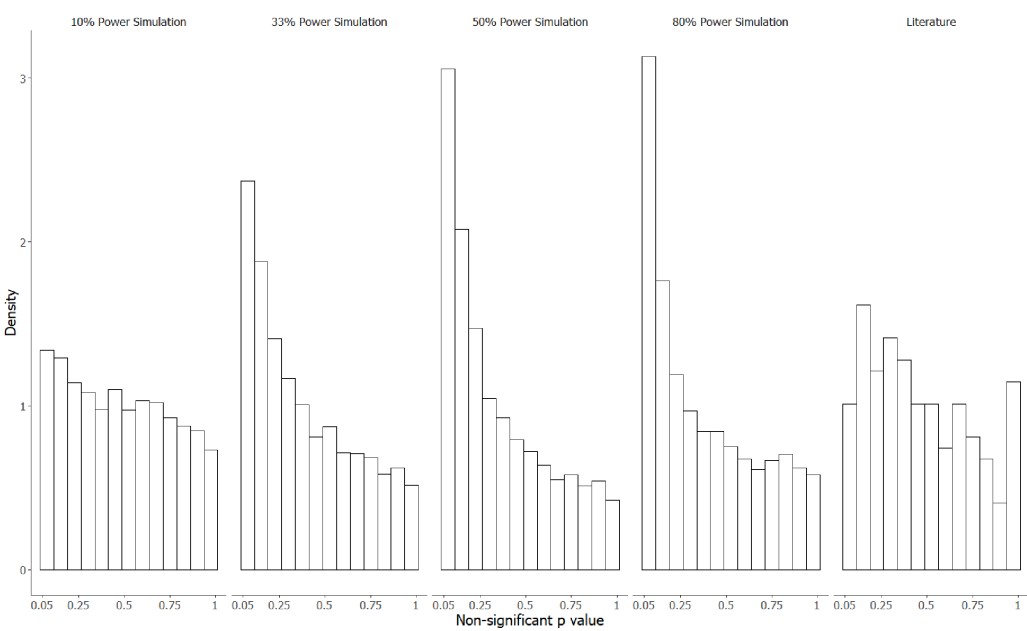

**Figure 3** **The observed *p*-value distribution of 198 *p*-values > .05, compared to three simulated distributions where 80% of alternative hypotheses were correct.** The observed *p*-value distribution was manually extracted from results corresponding to negative claims present in the abstracts of animal cognition articles. The observed *p*-value distribution was compared to three simulated distributions where 80% of alternative hypotheses were correct, with studies performed at either 10%, 33%, 50% or 80% statistical power.

## DISCUSSION

We extracted and classified how animal cognition researchers reported and interpreted the results of non-significant null hypothesis significance tests in 253 articles between 2019 and 2021. Across titles, abstracts, and results, we classified non-significant results as being reported with the "No Effect" phrasing that has often been labelled as erroneous in 84% of titles with a statistically non-significant result, in 63% of abstracts reporting a statistically non-significant result and in 41% of results sections reporting a statistically non-significant result. Reporting statistically non-significant results as "Non-Significant" was less common in titles and abstracts, but as prevalent as "No Effect" phrasings in the result sections (titles: 16%; reporting of results in abstract: 26%; result text: 52%). The other, albeit less frequently classified method of reporting statistically non-significant results was to comment on the similarity between groups or conditions (reporting of results in abstracts: 6%; result text: 4%).

Overall, these results demonstrate considerable heterogeneity in how animal cognition researchers report and interpret non-significant results in published articles. However, we often found it difficult to categorize results due to the heterogeneity in how statements referring to statistically non-significant results were phrased. Despite this heterogeneity, our results suggest that statistically non-significant results are at risk of being misreported and misinterpreted in animal cognition publications. It remains a question, however, what

the consequences of such misreporting might be, *i.e.*, how readers of scientific articles interpret "No Effect" statements, and this could be studied through analyzing how these studies are cited, in other publications but also in media reports and student essays.

A good example for the different ways in which the same phrasing can be interpreted by different researchers within the same research community are the three instances in which authors phrased the results of an ANOVA term as "no main effect", which we classified as "Ambiguous" as per our Coding guidelines. However, during the review process, one of the reviewers stated that they treat "no main effect" as formally equivalent to "no effect". The reason for our original classification was that referring to "no main effect" gives the reader more information about the statistical analysis used and thus may be more likely to be interpreted by readers correctly as the analysis yielding a non-significant main effect compared as when "no effect" is used. However, the reviewer's interpretation is as justifiable, and this example clearly illustrates the importance of investigating how researchers interpret and cite original findings in their own publications.

Possibly encouragingly, when researchers extended "No Effect" statements from reporting their study's results to interpreting them in relation to a substantive claim, they routinely opted for qualifiers to caveat inference to the populations (*e.g.*, "…these results *suggest* that there is no effect at the population level"). However, such qualified statements lack precision and are open multiple interpretations—they make only a vague suggestion that the strength of the evidence for the claim might be low, a claim that can often be explored in a quantitative and precise manner. Moreover, it is likely that such caveating is not unique to statistically non-significant results but also used to caveat significant findings, too. If that is correct, then the caveating may have more to do with researchers being critical of, or attempting to appear critical, of their results in general, and acknowledging that there may be alternative conceptual interpretations of their results (*Farrar & Ostojić, 2019*), rather than being specific to recognising the lack of information associated with studies with low power of statistical tests. However, more research is needed to pinpoint *precisely* how such statements are interpreted and implemented by scientists and the wider community. As already noted, one way in which researchers might reduce the ambiguity of their negative statements would be to use more formal methods of assessing evidence against informative null hypotheses, such as by testing against theoretically interesting effect sizes using as equivalence tests or comparing plausible null and alternative hypotheses using Bayes factors. Although beyond the scope of the current project, *Lakens (2017)* provides a detailed tutorial for equivalence testing in psychological research, and *Rose et al. (2018)* in animal behavior, and *Rouder et al. (2009)* provide an introduction to Bayes Factors. In addition, we would like to refer the reader to a competence model developed by *Edelsbrunner & Thurn (2020)* for all researchers who are involved in teaching statistics and mentoring students in the field of animal behavior science.

Notably, our coding team found it difficult to identify whether interpretations of a study's results in relation to a substantive claim in the abstracts were based on a statistically non-significant result and thus also to classify them in the next step. This difficulty likely reflects the distance between the theoretical claims researchers wish to test and the actual statistical hypotheses that are tested, *i.e.,* rarely can a theoretical prediction about an

animal's cognition be reduced to a single decision between a null and alternative hypothesis in a null hypothesis significance test.

Finally, we classified the formally incorrect reporting of results and interpretations of results as "No Effect" more commonly in abstracts and titles than "No Effect" reporting of results in the results section. That is, authors who have written out "Non-Significant" results in the results section nevertheless used the "No Effect" phrasing for reporting and interpreting the results in the abstracts and titles. This could be due to two factors, namely word limits and incentives to make bolder claims. If this is correct, then the former should be considered by journal editorial boards when setting their policy.

The $p$-value distribution likely differed from a uniform distribution for two reasons: the cumulative frequency was greater in the observed distribution for smaller $p$-values ($p < .3$) and was also greater for large $p$-values ($p > .95$). The larger density of smaller $p$-values is consistent with research with low-powered statistical tests in which the null hypothesis was incorrect, but which produces $p$-values that did not reach statistical significance. The density of very large $p$-values is consistent with researchers applying corrections that might increase $p$-values, such as Bonferroni corrections, or by using statistical tests with small sample sizes that produce non-uniform $p$-value distributions under the null hypothesis. An interesting contrast between the observed and simulated $p$-value distributions is that, unlike in the manual distribution, $p$-values in the range .05 to .10 were much more common than $p$-values in the range .10 to .15 in the simulated distributions. This is likely because we extracted results that we as coders had interpreted as being statistically non-significant for the manual dataset, but $p$-values in the range .05−0.1 are often interpreted by the original authors as "trends" or "marginally significant" and may therefore lead to author interpretations as if there had not been a statistically non-significant result.

## CONCLUSIONS

This study explored reporting and interpretation of statistically non-significant results in animal cognition literature through classification by other researchers in the field. In line with previous studies in other disciplines (*Aczel et al., 2018*; *Fidler et al., 2006*), we found that statistically non-significant results were often reported as if there were no differences observed between groups or conditions, and this was the case in the titles, abstracts and result sections of papers, although it was most frequent in the titles and abstracts. These results suggest that incorrect theoretical inferences based on non-significant results in animal cognition literature are common. However, because of the distance between statistical hypotheses and theoretical claims, and uncertainty around how no difference statements are interpreted, the consequences of this putative error are uncertain but may be grave. Nevertheless, these findings suggest that researchers should pay close attention to the evidence used to support claims of absence of effects in the animal cognition literature, and prospectively seek to, (i) report non-significant results clearly and formally correct, and (ii) use more formal methods of assessing the evidence against theoretical predictions.

## ACKNOWLEDGEMENTS

We would like to thank Balazs Aczel for discussions and clarifications about previous research in this area

### Funding

Benjamin G. Farrar was supported by the University of Cambridge BBSRC Doctoral Training Programme (BB/M011194/1). Alizée Vernouillet is currently supported by a BOF fellowship (BOF.PDO.2021.0035.01). Katharina F. Brecht was supported by a DFG Grant (BR 5908/1-1) and by a University of Tübingen Athene Fellowship. Mahmoud Elsherif is currently supported by The Baily Thomas Charitable Fund (TRUST/VC/AC/SG/5843-8995). Edward W. Legg is supported by a MSCA Fellowship (INTOM-794270). The funders had no role in study design, data collection and analysis, decision to publish, or preparation of the manuscript.

### Grant Disclosures

The following grant information was disclosed by the authors:
University of Cambridge BBSRC Doctoral Training Programme: BB/M011194/1.
BOF fellowship: BOF.PDO.2021.0035.01.
DFG Grant: BR 5908/1-1.
University of Tübingen Athene Fellowship.
The Baily Thomas Charitable Fund: TRUST/VC/AC/SG/5843-8995.
MSCA Fellowship: INTOM-794270.

### Competing Interests

Ljerka Ostojić is an Academic Editor for PeerJ.

### Author Contributions

- Benjamin G. Farrar conceived and designed the experiments, performed the experiments, analyzed the data, prepared figures and/or tables, authored or reviewed drafts of the article, and approved the final draft.
- Alizée Vernouillet performed the experiments, authored or reviewed drafts of the article, and approved the final draft.
- Elias Garcia-Pelegrin performed the experiments, authored or reviewed drafts of the article, and approved the final draft.
- Edward W. Legg performed the experiments, authored or reviewed drafts of the article, and approved the final draft.
- Katharina F. Brecht performed the experiments, authored or reviewed drafts of the article, and approved the final draft.
- Poppy J. Lambert performed the experiments, authored or reviewed drafts of the article, and approved the final draft.
- Mahmoud Elsherif performed the experiments, authored or reviewed drafts of the article, and approved the final draft.

- Shannon Francis performed the experiments, authored or reviewed drafts of the article, and approved the final draft.
- Laurie O'Neill performed the experiments, authored or reviewed drafts of the article, and approved the final draft.
- Nicola S. Clayton conceived and designed the experiments, authored or reviewed drafts of the article, and approved the final draft.
- Ljerka Ostojić conceived and designed the experiments, performed the experiments, authored or reviewed drafts of the article, and approved the final draft.

## Data Availability

The data are available at OSF: Farrar, Benjamin G. 2022. ''Non-Significant Results in Animal Cognition.'' OSF. October 30. osf.io/gdp6f.

## Supplemental Information

Supplemental information for this article can be found online at http://dx.doi.org/10.7717/peerj.14963#supplemental-information.

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
