# Peer review of "Reporting and interpreting non-significant results in animal cognition research"

_PeerJ, doi:10.7717/peerj.14963_

## Round 0.1 · original submission · Major Revisions

I was able to secure two comprehensive reviews of your manuscript. Both reviewers were quite positive about the paper, but also made several relevant comments that deserve author's attention. R1 mentions that it would be useful to see more background information in the intro with junior people in mind. I tend to agree with that. R2 points out two aspects about P-values (make them capital P througout the text) in the intro, and also suggests deepening the discussion on the implication of "no effect" result.

The organization of the text needs to be heavily reformulated.
In the introduction, avoid using footnotes and remove the subheadings, rearrange and integrate their content into the whole of the introduction to make it a single, coherent piece. The content of two out of four subheadings repeats what is already in the Methods, so remove them/

I'd also ask you to follow the PRISMA statement and provide a flow diagram detailing the data extraction and literature screening
http://www.prisma-statement.org/PRISMAStatement/FlowDiagram
Move the entire "Reliability and Quality Control" subheading in the Results to a suppl mat. It's not the main point of the paper and as presented, it breaks the reading. Focus on the findings of the survey in the Results.

·

Basic reporting

I feel like I’m missing a more detailed account of why certain statements/interpretations are incorrect and how researchers should describe various patterns of results. I think the paper would benefit from being written for a junior audience – much more detailed and explicit without assuming much background knowledge on the statistical and interpretive issues.
Apologies if I missed this, but I didn’t see information about the dates of publication that were covered in the article sampling process in the main text.
The authors are appropriately transparent with their process and the writing is generally clear.

Experimental design

The coding seems rigorous and I am pleased to see that some double-blind coding was done for reliability purposes as well as an accuracy review process.
I don’t understand why “no main effect of X” was considered different from “no effect of X.”
Line 275 indicates that 283 articles were analyzed but 253 articles were mentioned on line 403. Is this reconciled b/c 30 articles contained no null results?
I am impressed by the inclusive list of journals surveyed.

Validity of the findings

This is an important and relevant topic and has the potential to be an influential report. It is important to first understand how researchers are reporting null results and what errors are being made before presenting potential solutions so I commend the authors for taking this step.
I would strike “negative” on line 148 partly because I dislike the terms negative and positive when applied to findings, but also because I think the issue can apply to so-called “positive” results as well. For example, if researchers show no differences between training and transfer performance or between two groups. Even their footnote example is an example of the latter. It seems the authors use negative to refer to the lack of statistically significant differences, but this in itself is misleading given its broader use in the literature. The authors themselves should be more specific and precise with their language.
I agree with the authors that it will be interesting to examine how null effects are reported by other citing authors.

Additional comments

There are several minor errors:
Line 115 “interpreted” should be “interpretation.”
Line 127, is the word “articles” missing after the journal title?
Line 140 “no” should be “not.”
Keep punctuation inside “”
Line 157 should be “simulations.”
Line 242 should be “inter-rater”
Is line 389 an error?
Line 415 has an error.

·

Basic reporting

please see below

Experimental design

please see below

Validity of the findings

please see below

Additional comments

The authors present an empirical study reviewing interpretations of non-significant results in animal cognition research.

I appreciate the idea behind this work and think that the article is well-written. I am not an expert in animal cognition research, but I have authored work on misinterpretations of non-significant results. I am also not a statistician, but I believe that I can make some suggestions to further improve the quality of the introduction, and also of some aspects of the reporting in the method.

Introduction:
l. 70-71: I don't understand how researchers can interpret that there was no effect in a sample. A p-value is clearly and always an inferential statistic that is about a population, not about a sample. You might want to clarify what you mean by this in contrast to the "target population in general" mentioned in the second part of the sentence.

l. 126-143 I appreciate this part a lot.

l. 200: Same as my first point - If authors say that non-significant results arise "in a sample", ok - but a p-value just doesn't tell us anything about the sample - it's actually completely devoid of any sample information. It's a percentage of hypothetical studies in a hypothetical distribution - thus although p is computed based on data, I have a really hard time seeing how it can describe a characteristic of the sample and not the population.

l. 278-313: I appreciate the detalined report on rating difficulties - this is so useful!

l. 297-304: For any numbers where this is applicable, please report a proper interrater-reliability statistics such as kappa or Gwet's AC1 (or, even better, both), instead of raw percentage agreement. These statistics correct for chance agreement. I think this is applicable to, and should be done with, your classification-ratings.

Quality control: I am not sure why this is different to interrater-reliability. Could you please elaborate on the difference, and, again, report proper IRR-statistics where applicable?

l. 389: "Error! Reference source not found" :D

Discussion:
I agree with most of the discussion.
Edelsbrunner & Thurn (preprint, https://psyarxiv.com/j93a2/) had the idea to code researchers' own inferences that base on misinterpretations of non-significant results. Did you consider doing this? Why suggest that this is up to readers, when authors commonly draw inferences based on such misinterpretations themselves directly in their articles' discussions?
Please consider discussing a bit more the implications of your finding of the common "no effect"-misinterpretation. Simply stating that it is unclear how authors mean this and how readers interpret it is not only unsatisfying, but I do actually believe that the reported mistaken interpretations are rather severe. For example l. 422: "these results suggest that there is no effect at the population level" is a clearly wrong theoretical inference - which is clearly among the more severe implications that a misinterpreted non-significant effect can have. This is just my view on this, however, and if the authors disagree, I would nonetheless ask them to expand the discussion around lines 410 - 430.

Kind regards

Signed
Peter Edelsbrunner
ETH Zurich

---

## Round 0.2 · accepted · Accept

Thank you for preparing a carefully revised version of the manuscript. I have now received the comments from the same two reviewers from the previous round and they have no further concerns. One of them pointed out a few minor adjustments that could easily be incorporated in the proof stage

·

Basic reporting

The paper is very clear and thoughtful.

Experimental design

I have no further comments about the design of the study.

Validity of the findings

The conclusions are clear and follow directly from the coding.

Additional comments

Line 42, “issues” should be “issue”
Lines 67-69, there is a weird gap in spacing midway through the sentence in the proof
On line 110, does it make sense to state that the p value is really the probability of obtaining the given set of results IF the null hypothesis is true?
Line 238, 257 and check elsewhere, place punctuation inside of quotations.
Line 357 delete “as”

·

Basic reporting

All is good now

Experimental design

All is good now

Validity of the findings

All is good now

Additional comments

The authors have very nicely answered and integrated my points.
Thank you!